# AURELIUS: RELATION AWARE TEXT-TO-AUDIO GENERATION AT SCALE

**Yuhang He**[†]  **He Liang**[‡]  **Yash Jain**[†]  **Andrew Markham**[‡]  **Vibhav Vineet**[†]

[†] Microsoft Research    [‡] Department of Computer Science, University of Oxford

✉  Corresponding: `yuhanghe@microsoft.com`

68  Code: https://github.com/yuhanghe01/Aurelius

ⓔ  Project: https://yuhanghe01.github.io/Aurelius-Proj/

## ABSTRACT

We present *Aurelius*, a new framework that enables relation aware text-to-audio (TTA) generation research at scale. Given the lack of essential audio event and relation corpora, *Aurelius* contributes a large-scale audio event corpus *AudioEventSet* and another large-scale relation corpus *AudioRelSet*. Comprising 110 event categories, *AudioEventSet* maximally covers all commonly heard audio events and each event is unique, realistic and of high-quality. *AudioRelSet* consists of 100 relations, comprehensively covering the relations that present in the physical world or can be neatly described by text. As the two corpora provide audio event and relation independently, they can be combined to create massive `<text,audio>` pairs with our pair generation strategy to support relation aware TTA investigation at scale. We comprehensively benchmark all existing TTA models from both general and relation aware evaluation perspective. We further provide an in-depth investigation into scaling existing TTA models' relation aware generation by either training from scratch or leveraging cross-domain general TTA knowledge. The introduced corpora and the findings from investigation potentially facilitate future research on relation aware TTA generation.

## 1 INTRODUCTION

Text-to-audio (hereinafter TTA) generation task aims at generating acoustically high-fidelity audio whose content is inferred from the input text. Owing to the success of generative modeling (*e.g.*, diffusion based (Ho et al., 2020; Xue et al., 2024), score based (Vahdat et al., 2021) and flow matching based (Lipman et al., 2023; Guan et al., 2024) methods) and the availability of large `<text,audio>` pair dataset (*e.g.*, AudioCaps (Kim et al., 2019), AudioSet (Gemmeke et al., 2017)), we have witnessed significant advancement in general TTA task in recent years (Ghosal et al., 2023; Hung et al., 2026; Liu et al., 2024). Despite these achievements, the relation aware TTA generation still remains as a challenging task as it jointly requires audio event generation and relation modeling. Audio events and their relation are two fundamental elements humans rely on for holistic acoustic scene understanding or engaging communication (Zacks et al., 2007; Hirsh et al., 1967; Lake et al., 2015). We humans can interpret the relation and audio events within the textual description with ease to decide how the target audio looks like. Enabling TTA models with similar relational reasoning and event interpretation capability is therefore essential for bridging the gap between relation aware TTA model quality and human-level crossmodal reasoning.

The recent preliminary investigation by RiTTA (He et al., 2025) already shows the incapability of existing TTA models in relation aware generation, but the investigation runs on top of small relation and audio event corpora. The data corpora small scale issue naturally hinders further investigation. To enable relation aware TTA at scale, we introduce *Aurelius*, a novel framework that contributes to relation aware TTA from both dataset benchmark and technical methodology aspects. From the dataset benchmark aspect, we meticulously curate two large-scale corpora: *AudioEventSet* and *AudioRelSet*. *AudioEventSet* is an audio event corpus that comprises 110 across fine-grained event classes across 7 main acoustic categories we commonly hear in our daily lives. In contrast to existing audio event datasets (Gemmeke et al., 2017; Kim et al., 2019; Fonseca et al., 2022) that are either

noisy, polyphonic or label-missing, *AudioEventSet* provides a coarse-to-fine tree structured audio event corpus that is both internally distinctive and externally comprehensive. Each individual audio event in *AudioEventSet* is high-quality, realistic and intra-class diverse. *AudioRelSet* is the large-scale relation corpus with up to 100 detailed relations completely covering the potential relations audio events may present in the 3D physical world or text can describe succinctly. *AudioRelSet* is also tree structured and can be further scaled up to incorporate more relations. Each relation in *AudioRelSet* has an "arity" property that is further used to combine relation and audio events together to create `<text,audio>` pairs for relation aware TTA task. *AudioEventSet* and *AudioRelSet* are orders of magnitude larger than existing relevant dataset, enabling thorough and in-depth investigation for relation aware TTA task.

Based on the introduced audio event corpus *AudioEventSet* and relation corpus *AudioRelSet*, we further introduce a `<text,audio>` pair generation strategy that can generate essential `<text,audio>` pairs with both audio event based and textual description diversity. As the audio event corpus is disentangled from relation corpus, our proposed strategy can generate nearly unlimited `<text,audio>` pairs tailored for various training requirements. In summary, as illustrated in Fig. 1, *Aurelius* advances relation aware TTA research by contributing large-scale corpora of audio events and relations, together with a dedicated framework for relation aware generation. The explicit disentanglement of audio events and relations, the hierarchical tree-structured design of each corpus, and the

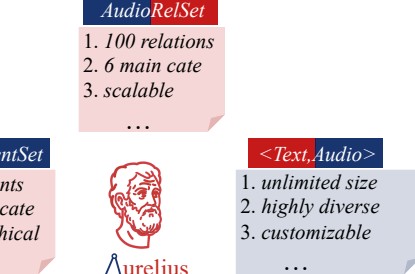

Figure 1: Aurelius contributes to relation aware TTA by introducing an audio event corpus *AudioEventSet*, a relation corpus *AudioRelSet* and `<text,audio>` pair generation strategy.

systematic `<text,audio>` creation strategy collectively provide a strong foundation for curating essential datasets in this domain. Building on this foundation, our proposed *AudioRelGen* framework tackles relation aware TTA by decoupling audio event modeling from relation modeling, offering an essential first step toward structured audio generation. We believe this work will not only establish a new benchmark for relation aware TTA but also inspire future research on modeling complex event–relation dynamics in sound.

## 2 RELATED WORK

**Text-to-Audio Generation** aims at generating the audio waveform that semantically aligns well with the input text. The fast development of generative modeling techniques (Ho et al., 2020; Vahdat et al., 2021; Lipman et al., 2023) in recent years has largely advanced the TTA generation in terms of high-fidelity and high-intelligibility (Liu et al., 2024; 2023; Kreuk et al., 2023; Yang et al., 2022; Ghosal et al., 2023; Liao et al., 2024), alongside other crossmodal generation tasks including but not limited to text-to-music (TTM, *e.g.*, MusicGen (Copet et al., 2023) and MusicLM (Agostinelli et al., 2023)), image-to-audio (I2A, *e.g.*, RegNet (Chen et al., 2020), Img2Wav (Sheffer & Adi, 2023) and SpecVQGAN (Iashin & Rahtu, 2021) and text-to-image (T2I). Although the promising achievement in generating realistic and semantically text-aligned audio, existing TTA methods still perform poorly in relation aware TTA generation. Prior work like RiTTA (He et al., 2025) and CompA (Ghosh et al., 2024) have preliminarily explored relation aware TTA and shown the incapability of existing TTA methods through limited audio event and relation corpora, which inevitably hinders future investigation at scale. Moreover, publicly available audio event corpora (AudioSet) are directly collected from either online video data or audio sharing platform without proper quality check, resulting in audio events being label-missing, noisy and ambiguous. Our work circumvents these barriers by introducing a meticulously curated audio event corpus *AudioEventSet* that is high-quality, distinctive and realistic, potentially covering all commonly heard audio events.

**Relation Modeling** has been widely discussed within modalities, including images (Liu et al., 2022; Zerroug et al., 2022), natural language processing (Wadhwa et al., 2023), and acoustics (Xie et al., 2025a; Ghosh et al., 2024; He et al., 2025). In the context of 2D images, the objects of interest can exhibit compositional and spatial relations (Liu et al., 2022; Zerroug et al., 2022). In the context of the

3D physical world, an audio event is the most fundamental acoustic signal, and multiple audio events join together to represent the 3D physical world via more sophisticated relations than image-based relations, ranging from basic spatial, temporal, and perceptual relations to their nested combinations. Prior works (Xie et al., 2025a; Ghosh et al., 2024; He et al., 2025; Xie et al., 2025b) have discussed audio-event relations at small scale and with minimal complexity, making them hard to scale up to accommodate the potential relation complexity present in either the 3D physical environment or textual descriptions. To fill this gap, we curate *AudioRelSet*, a large-scale relation corpus that reflects the relations potentially present in the physical world and can be neatly described by text.

**Text-to-Audio Generation Techniques.** Existing TTA methods can be divided into two main categories: early methods are diffusion based (Liu et al., 2024; 2023; Kreuk et al., 2023; Yang et al., 2022; Ghosal et al., 2023; Liao et al., 2024; Xue et al., 2024), while recent methods are flow-matching based (He et al., 2025; Hung et al., 2026; Guan et al., 2024). Flow-matching methods are usually faster during both training and inference and can yield better performance than diffusion-based methods. We comprehensively benchmark these methods on our introduced corpora and further provide an in-depth investigation to reveal potential ways to scale up existing TTA methods' relation aware capability.

## 3 AURELIUS BENCHMARK: AUDIOEVENTSET AND AUDIORELSET

### 3.1 AUDIO EVENT CORPUS: *AudioEventSet*

An audio event refers to an auditory signal occurring over a specific period of time, typically representing an independent, human-recognizable sound. To support relation aware TTA research, the desired audio-event corpus should be: 1. diverse enough to maximally accommodate the wide variety of audio events potentially present in the 3D physical world; 2. clean and high-fidelity to enable reliable in-depth technical investigation; 3. distinctive so that events can be easily distinguished without ambiguity; 4. hierarchically organized with respect to genre to enable investigation at different granularity. After thorough investigation of existing audio-event datasets, however, we

Table 1: Audio Event Dataset Comparison.

| Dataset | Characteristic |
|---|---|
| AudioSet (2017) FSD50K (2022) AudioCaps (2019) AudioTime (2025a) | *polyphonic, ambiguous, noisy, label-missing* |
| *AudioEventSet* | *distinctive, high-quality, clean, hierarchical coarse-to-fine intra-class diversity inter-class discriminative* |

find that all existing datasets fall short of these four properties. As shown in Table 1, existing audio-event datasets (*e.g.*, AudioSet (Gemmeke et al., 2017), AudioCaps (Kim et al., 2019), AudioTime (Xie et al., 2025a), and FSD50K (Fonseca et al., 2022)) are noisy, label-missing, polyphonic (multiple events temporally overlap), or semantically ambiguous (multiple event classes correspond to the same audio). To address this dilemma, we introduce *AudioEventSet*, a meticulously curated audio-event corpus that is intrinsically clean, diverse, distinctive, and hierarchically organized.

*AudioEventSet* ontology is tree-structured with depth three. From the root node to the leaf node, each audio event is organized in a coarse-to-fine granularity. As shown in Fig. 2 and Table I in the Appendix, we build on RiTTA (He et al., 2025) to categorize *AudioEventSet* into 7 main categories: five singular-source categories *Animal*, *Human*, *Machinery*, *Music*, and *Nature*, and two interaction-based categories, *Human-Object* and *Object-Object* interactions. The seven categories maximally cover commonly heard audio events in the 3D physical world. Each main category is associated with multiple subcategories, each of which is further associated with multiple fine-grained event classes. For example, the *Human* main category contains *human voice*, *human speech*, *hands action*, *group action*, and *locomotion* subcategories, comprehensively categorizing human-centered audio events from various aspects.

During *AudioEventSet* ontology construction, we guarantee each curated audio event is distinctive, unique, and human-distinguishable. An audio event emitting ambiguous or nondistinctive audio is discarded. For example, *engine idling* in AudioSet (Gemmeke et al., 2017) audio differs significantly across engines, and it is easily confused with other audio events such as *working fan* and *hairdryer*. We thus exclude all of them from the corpus. Moreover, we account for audio event source origin, event

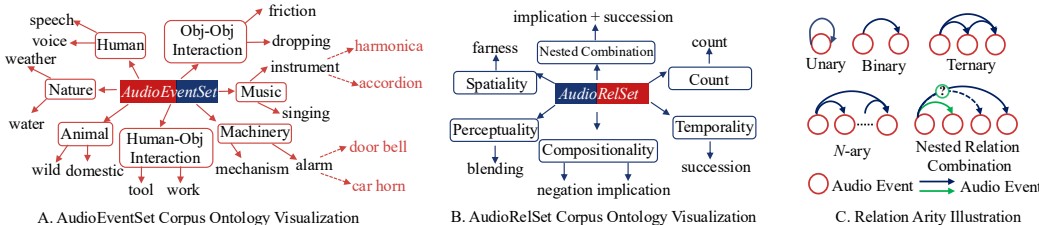

A. AudioEventSet Corpus Ontology Visualization  B. AudioRelSet Corpus Ontology Visualization  C. Relation Arity Illustration

Figure 2: *AudioEventSet* and *AudioRelSet* corpora illustration: we visualize the *AudioEventSet* ontology in sub-figure A. It is tree-structured with depth 3 and contains 7 main categories and 110 event categories (leaf nodes) in total. We show only part of the leaf nodes (with a red dotted arrow) for clarity. The detailed event ontology is given in Table I in the Appendix. The *AudioRelSet* ontology is shown in sub-figure B; it is tree-structured with depth 2 and contains 6 main categories and 100 categories in total. The detailed relation ontology is given in Table II in the Appendix. In sub-figure C, we conceptually illustrate the relation "arity", which is used to connect relations and audio events to generate audio.

category, and the physical mechanisms of audio generation in *AudioEventSet* ontology construction. For example, in the Object-Object main category, we exhaustively consider impact, friction, dropping, and explosion mechanisms. In summary, we have curated 110 audio events, which is four times larger than the audio-event corpus proposed in RiTTA (He et al., 2025); each leaf audio event is associated with around 75 realistic audio snippets ranging from 1 s to 5 s.

For each leaf-node audio event, we collect exemplar audio from either the copyright-free `freesound.org` platform or FSD50K (Fonseca et al., 2022). As most audio from `freesound.org` and FSD50K [1] are real recordings shared by volunteers across the globe, the collected audio for each event is diverse and realistic enough to reflect audio events in the physical world. Manual verification is adopted to ensure content correctness and label consistency. We argue that the curated *AudioEventSet* can be applied to tasks beyond TTA, and we anticipate much wider usage of the dataset.

## 3.2 RELATION CORPUS: *AudioRelSet*   *AudioRelSet*

Prior works (Xie et al., 2025a; He et al., 2025; Ghosh et al., 2024) have explored audio-event relations from various perspectives, but only at small scale. For example, AudioTime (Xie et al., 2025a) and CompA (Ghosh et al., 2024) discuss temporal relations. RiTTA (He et al., 2025) additionally introduces spatial, compositional, and count relations, resulting in a total of 11 relations. In this section, we introduce *AudioRelSet*, a meticulously curated large-scale relation corpus with up to 100 distinct relations. To ensure *AudioRelSet* exhibits real-scenario practicability, text-manageable complexity, and relation scalability, we follow three guidelines to curate *AudioRelSet*: 1. maximally cover the potential relations audio events can present in the 3D physical world; 2. include sufficient relation complexity while still being efficiently and neatly described by text; 3. allow the relation corpus to scale up to accommodate more sophisticated relations. To this end, we construct six fundamental relation categories, in which four main relations describe relations present in the 3D physical world, one focuses on a TTA model's logical reasoning capability, and the last derives from nested combinations of the five main relations.

*AudioRelSet* ontology is tree-structured with depth 2. The root node connects six main relations, each of which further associates with multiple sub-relations. Let $\mathcal{E} = \{E_1, E_2, \ldots, E_m\}$ denote the audio events in *AudioEventSet* introduced in Sec. 3.1, and let $\mathcal{R} = \{R_1, R_2, \ldots, R_n\}$ denote the relations to be constructed. *AudioRelSet* is represented as follows:

1. **Temporality** describes the sequence or overlap of audio events in the time domain; it contains four sub-relations: *Precedence*: $E_1 \prec E_2$ (event $E_1$ occurs before $E_2$); *Succession*: $E_1 \succ E_2$ (event $E_1$ occurs after $E_2$); *Simultaneity*: $E_1 \parallel E_2$ ($E_1$ and $E_2$ occur concurrently); *Repetitiveness*: $\sim E_1$ (event $E_1$ occurs repetitively in the time domain).

---

[1]FSD50K (Fonseca et al., 2022) data is also sourced from `freesound.org`

2. **Spatiality** defines the relative spatial positions or motion status between or within audio events; it contains five sub-relations: *Proximity*: $d(E_1, E_2) \leqslant \tau$ ($E_1$ and $E_2$ are within distance $\tau$); *Closeness*: $d(E_1) < d(E_2)$ ($E_1$ is closer than $E_2$); *Farness*: $d(E_1) > d(E_2)$ ($E_1$ is farther than $E_2$); *Approaching*: $\frac{d}{dt}d_{E_1}(t) < 0$ ($E_1$ is moving closer); *Departing*: $\frac{d}{dt}d_{E_1}(t) > 0$ ($E_1$ is moving away).

3. **Count** focuses on the number of audio events that take place within a period of time: *Count*: $|\mathcal{E}| = N, N \in \mathbb{Z}^+$. (the cardinality of $\mathcal{E}$ is the number).

4. **Perceptuality** introduces six acoustic effects to an audio event,

- *Balancing*: $\mathcal{R}_{\text{balance}}(E_1, E_2, \sigma)$ (level balance between $E_1$ and $E_2$ by balancing factor $\sigma$, so that one event dominates and the other serves as the background audio).
- *Blending*: $\mathcal{R}_{\text{blend}}(E_1, E_2, \theta)$ (mix $E_1$ and $E_2$ together by factor $\theta$ so as to be indistinguishable).
- *Reverberation*: $\mathcal{R}_{\text{reverb}}(E_1)$ applies a reverberation effect to $E_1$, as if it is heard in a canyon.
- *Time-stretching*: $\mathcal{R}_{\text{stretch}}(E_1, \alpha)$, where $\alpha$ is the time-stretching factor and $E_1$ sounds slower.
- *Amplification*: $\mathcal{R}_{\text{amp}}(E_1, \beta)$, where $\beta$ is the amplification factor and $E_1$ sounds louder.
- *Attenuation*: $\mathcal{R}_{\text{att}}(E_1, \gamma)$, where $\gamma$ is the attenuation factor and $E_1$ sounds quieter.

5. **Compositionality** indicates the logical operations within audio events that TTA models need to reason about before deciding what audio events to generate. It contains five sub-relations.

- *Conjunction*: $E_1 \wedge E_2$ (both events occur).
- *Disjunction*: $E_1 \vee E_2$ (at least one event occurs, or both occur).
- *Negation*: $\neg E_1$ (the absence of the event $E_1$ in the generated audio).
- *Exclusive Or*: $(E_1 \vee E_2) \wedge \neg(E_1 \wedge E_2)$ (either $E_1$ or $E_2$ occur, but not both).
- *Implication*: $E_1 \Rightarrow E_2, \neg E_1 \Rightarrow E_3$ (if $E_1$ occurs, then $E_2$ occurs; otherwise, $E_3$ occurs).

6. **Nested Combination** is a hierarchical structuring of multiple basic relations (*e.g.*, the aforementioned *Temporality*, *Spatiality*), such that the output of one relation serves as the input or context for another, forming a directed acyclic relation structure. Nested combination allows for capturing complex relation interactions among audio events. For example, by nesting *Implication*, *Approaching* and *Conjunction*, we can generate a more complex text prompt showing below,

> **Nest Combination Example:** *Implication*, *Approaching* and *Conjunction*
>
> *If* generated both {A} event and {B} event, → *Conjunction*
> *then* continue to generate {C} audio event,
> *else* just generate {D} audio event that is gradually approaching close. →*Approaching*

Mathematically, the relation $R_{\text{nested}}(E)$ resulting from nested combination can be represented as,

$$R_{\text{nested}}(E) = R_n(R_{n-1}(\dots R_2(R_1(E))\dots)) \tag{1}$$

where $E = \{e_1, e_2, \dots, e_m\}$ represents a finite set of audio events. We combine relations arising from the introduced 5 basic relations to construct nested combination relations and have created 79 nested combination relations.

It is worth noting that nested combinations are scalable, and we can theoretically construct more complex nested relations (even infinite relations) by involving more basic relations in the nesting process. In this work, we constrain nested combinations to involve at most five audio events (*Quinary*). It remains a future research topic to explore more complex nested

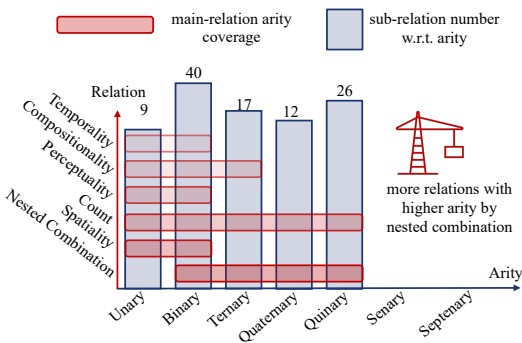

Figure 3: Arity coverage in *AudioRelSet*.

combinations, and the key challenge is how to construct concise and precise textual descriptions for highly complex nested relations. Moreover, during the nesting process, we explicitly run internal logical-correctness and feasibility checks before accepting a nested relation; any nested relation that violates these rules is abandoned. For example, the combination of *Count* and *Conjunction* internally equals *Count*.

**Relation Arity.** each relation in *AudioRelSet* is associated with an "arity" property, which indicates the audio event number it requires to represent the relation. The visual illustration of arity is shown in Fig. 2 C. The arity coverage across *AudioRelSet* main relation categories is given in Fig, 3, from which we can see that the arity ranges from 1 to 5 (unary to quinary) and most main relation cuts across multiple arities. Moreover, the construction of more complex relations introduces higher arity. We use "arity" to create <text,audio> pairs (see Sec. 3.3) and experiment evaluation (see Sec. 4).

## 3.3 TEXT-AUDIO PAIR CREATION: *<Text, Audio>*

With the constructed audio event corpus in Sec. 3.1 and relation corpus in Sec. 3.2, we can further construct relation aware <text,audio> pairs. Specifically, as is shown in Fig. 4, we first associate each of the 100 relations in the relation corpus with meticulously curated 5 text description templates. We either manually write or query GPT-4o to generate 5 text prompt templates precisely describing the relation and accommodating the large language usage variation (see Fig. 4 line 4-8). Each template contains audio events name placeholder, we instantiate the template by replacing the placeholder with real audio event name to obtain the text prompt. To accommodate the synonymy of audio event name, we maintain a synonym list for each audio event name, and randomly select one each time when instantiating the template. For example, the audio event name "hammer nailing" can be synonymously replaced by one of [*hitting*, *slapping*, *smacking*, *punching*].

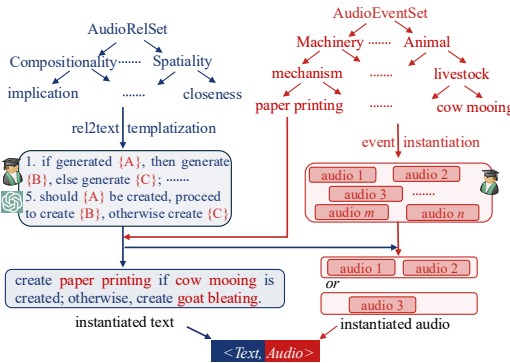

Figure 4: <text,audio> pair generation illustration, which can generate nearly unlimited pairs with high diversity.

To accurately describe the audio event with text, we adopt the "Head-Modifier Structure with Progressive Verb Form" approach. In this approach, the description begins with the subject or entity producing the audio (*e.g.*, "food") as the head, emphasizing the primary source of the sound. The action is then specified using its present participle form (*e.g.*, "frying") as the modifier to convey a sense of immediacy and highlight that the audio event is ongoing. For instance, instead of describing a sound as "frying food" or "fry food" it is labeled as "food frying audio," where the subject ("food") is foregrounded, and the action ("frying") contextualizes the nature of the audio. This approach ensures clarity, aligns with the temporal context of the audio, and effectively captures the dynamic nature of the event. With the same audio events name, we can retrieve its relevant audio waveform data and generate the audio by following the relation (He et al., 2025).

## 4 EXPERIMENT

### 4.1 DATASET CONSTRUCTION

Following the common setup in existing TTA model, the created audio is 10 second long with sampling rate 16 kHz. Based on the data creation method introduced in Sec. 3.3, in the training phase we randomly construct 360 <text,audio> pairs for each relation, resulting in 36,000 pairs in total. In the testing phase, we randomly construct 100 <text,audio> pairs for each relation, ensuring that no constructed pairs appear in the training dataset. Following prior TTA model settings, we set each audio sample to 10 seconds with a 16 kHz sampling rate; the training dataset is 100 hours, and the testing dataset is 28 hours. Since we decouple relations from audio events during dataset

construction, and the training texts differ from the testing texts, the constructed training and testing `<text,audio>` pairs have no overlap and differ significantly.

## 4.2 Evaluation Metric

We use both classic general evaluation metrics and relation aware evaluation metrics. For general evaluation, we follow traditional TTA works (Liu et al., 2024; 2023; Ghosal et al., 2023; Majumder et al., 2024) and adopt three metrics: Fréchet Audio Distance (FAD), Fréchet Distance (FD) (Heusel et al., 2017), and Kullback–Leibler (KL) divergence. These three metrics measure the overall similarity in embedding space between reference audio and generated target audio without explicitly taking relations into account. Specifically, following prior TTA works, we extract embeddings from the VGGish (Hershey et al., 2017) model for FAD and KL, and embeddings from the PANNs (Kong et al., 2020) model for FD.

For relation aware evaluation, we adopt the multi-stage relation aware (*MSR*) evaluation protocol introduced in RiTTA (He et al., 2025). In the *MSR* protocol, we first explicitly extract audio events and relations $(E', R')$ from generated audio, and then compare them with reference audio events and relations $(E, R)$. To reflect whether the model has generated, and only generated, the designated audio events and relations, *MSR* adopts *Pre*sence, *Rel*ation correctness, and *Par*simony scores to gauge the quality of generated audio from different perspectives. Specifically, we report `mAPre`, `mARel`, and `mAPar` scores for either individual relations or all relations. More detailed information about the *MSR* metric can be found in RiTTA (He et al., 2025). To extract audio events from generated audio, we fine-tune an audio event detection and tagging model on top of the pre-trained PANNs (Kong et al., 2020) model with a 1-million-sample training dataset. The mAP on a 100,000-sample testing dataset reaches 0.91 for audio event detection, ensuring the fine-tuned model can extract all potential audio events with high precision. To classify acoustic effects, we train another seven-class acoustic-effects classifier on top of the pre-trained PANNs model with a 1-million-sample training dataset. The accuracy on a 100 k testing dataset reaches 95%.

## 4.3 Benchmarking Methods

We exhaustively benchmark 9 of the most recent general TTA models: AudioLDM (Liu et al., 2023), AudioLDM 2 (Liu et al., 2024), MakeAnAudio (Huang et al., 2023), AudioGen (Kreuk et al., 2023), Tango (Ghosal et al., 2023), Tango 2 Majumder et al. (2024), LAFMA (Guan et al., 2024), Auffusion (Xue et al., 2024), and TangoFlux (Hung et al., 2026). These models are pre-trained on general TTA datasets (Gemmeke et al., 2017; Kim et al., 2019). For benchmarking, we use their released checkpoints to generate 10 second audio from text prompts; detailed configurations are provided in Table III in the Appendix.

We further benchmark two agentic workflow based methods, in which we leverage an open-source Qwen-family LLM as an agent to analyze the input text and output the separate audio events a TTA model needs to generate. The purpose of experimenting with this agentic workflow is to see whether we can decompose the relation aware generation task into simpler single-audio-event generation tasks. Detailed implementation of the agentic workflow is provided in Appendix .3.

## 4.4 Benchmarking Result on Existing TTA Models

The benchmarking results are shown in Table 2, from which we observe that all existing TTA models perform poorly on relation aware TTA generation. Similar to RiTTA (He et al., 2025), we also find a contradictory evaluation outcome between general evaluation and relation-aware evaluation, which highlights the specificity of the relation-aware TTA task. Among all benchmarking methods, Audio-Gen (Kreuk et al., 2023) and TangoFlux (Hung et al., 2026) perform best. While AudioGen (Kreuk et al., 2023) achieves the best mAPar (relation parsimony) and mAMSR, TangoFlux (Hung et al., 2026) is best in mAPre and mARel, which means it excels at accurately generating the target audio events and corresponding relations. However, almost all benchmarking methods achieve less than a 10% accuracy rate across all relation aware evaluation metrics, which in turn verifies the need to introduce a new large-scale benchmark tailored for relation aware TTA research.

Furthermore, both agentic-flow baselines perform poorly; they are substantially worse than most existing TTA approaches. This poor performance highlights a critical limitation: simply scaling up

Table 2: Quantitative benchmarking result on our introduced benchmark. mAPre, mARel and mAPar are in $10^{-2}$. mAPre and mARel can be treated as *presence*, *relation correctness* percentage ratio, they lie in range $[0, 100]$. mAPar score also lies within $[0, 100]$. mAMSR (%) lies in range $[0, 1]$

| Eval Way | Model | #Param | General Evaluation | | | Relation Aware Evaluation %(↑) | | | |
|---|---|---|---|---|---|---|---|---|---|
| | | | FAD ↓ | KL ↓ | FD ↓ | mAPre | mARel | mAPar | mAMSR |
| Zero-Shot | AudioLDM (s-full) 2023 | 185 M | 4.02 | 21.23 | 22.36 | 3.47 | 0.91 | 2.95 | 0.73 |
| | AudioLDM (l-full) 2023 | 739 M | 4.13 | 22.05 | 23.03 | 3.10 | 0.79 | 2.63 | 0.63 |
| | AudioLDM 2 (l-full) 2024 | 844 M | 4.54 | 22.90 | 30.53 | 0.35 | 0.04 | 0.31 | 0.03 |
| | MakeAnAudio 2023 | 452 M | 5.10 | 50.97 | 30.49 | 4.75 | 0.88 | 4.05 | 0.73 |
| | AudioGen 2023 | 1.5 B | 7.97 | 25.19 | 32.29 | 11.3 | 2.84 | 9.13 | 2.22 |
| | LAFMA 2024 | 272 M | 25.85 | 269.54 | 65.27 | 0.96 | 0.15 | 0.45 | 0.07 |
| | Auffusion 2024 | 1.1 B | 4.13 | 42.59 | 31.17 | 6.71 | 1.41 | 4.07 | 0.79 |
| | Tango 2023 | 866 M | 7.47 | 64.10 | 28.28 | 4.46 | 0.98 | 3.67 | 0.79 |
| | Tango 2 2024 | 866 M | 9.59 | 65.24 | 35.50 | 9.68 | 2.48 | 5.49 | 1.29 |
| | TangoFlux 2026 | 576 M | 6.01 | 26.73 | 30.00 | 12.38 | 3.34 | 7.28 | 1.77 |
| Agentic | Qwen2 7B+TangoFlux | - | 9.98 | 142.87 | 39.20 | 3.53 | 0.77 | 2.25 | 0.04 |
| | Qwen2.5 32B+TangoFlux | - | 9.70 | 140.56 | 38.65 | 3.79 | 0.96 | 2.41 | 0.60 |

current TTA methods without fundamentally enhancing their relation aware modeling capability is unlikely to succeed. In this light, the benchmark introduced in this paper is not merely a comparison tool but a catalyst, providing the necessary structure, evaluation, and motivation to drive genuine advances in relation aware TTA research.

## 4.5 TWO INTUITIVE WAYS TO IMPROVE RELATION AWARE MODELING

Table 3: Quantitative result comparison on testset between finetuning and training from scratch (scratch) on curated 100 hours dataset.

| Train Strategy | Model | #Param | General Evaluation | | | Relation Aware Evaluation %(↑) | | | |
|---|---|---|---|---|---|---|---|---|---|
| | | | FAD ↓ | KL ↓ | FD ↓ | mAPre | mARel | mAPar | mAMSR |
| finetuning | Tango 2023 | 866 M | 3.88 | 33.26 | 21.30 | 14.58 | 4.18 | 10.16 | 2.73 |
| | Tango 2 2024 | 866 M | 4.06 | 22.39 | 20.32 | 15.53 | 4.63 | 10.21 | 2.86 |
| | TangoFlux 2026 | 576 M | 1.29 | 9.68 | 16.44 | 28.57 | 8.02 | 20.84 | 5.58 |
| scratch | Tango 2023 | 866 M | 3.63 | 22.34 | 20.16 | 14.89 | 3.69 | 10.98 | 2.64 |
| | TangoFlux 2026 | 576 M | 1.64 | 17.82 | 11.72 | 16.68 | 3.82 | 12.01 | 2.58 |

Two intuitive strategies to enhance relation aware modeling in existing TTA methods are (i) fine-tuning on our curated dataset and (ii) training from scratch. This dual perspective not only tests the feasibility of our benchmark but also evaluates the potential to transfer general TTA-domain knowledge into relation-aware settings. To this end, we apply both training strategies to three representative baselines: Tango (Ghosal et al., 2023), Tango 2 (Majumder et al., 2024), and TangoFlux (Hung et al., 2026). The results in Table 3 reveal a clear trend: both fine-tuning and training from scratch substantially improve relation-aware performance, validating the effectiveness of our benchmark as a testing ground for relation-aware TTA. Notably, TangoFlux benefits the most from fine-tuning, indicating that cross-domain TTA knowledge can be effectively transferred to relation aware tasks. In contrast, Tango shows little difference between the two strategies, suggesting that model architecture and inductive bias may

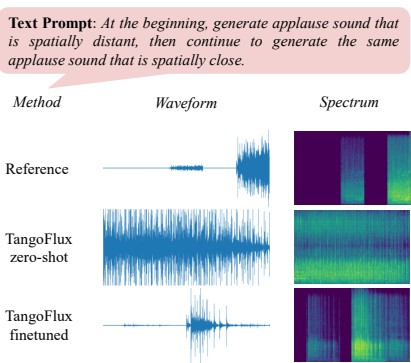

Figure 5: Qualitative comparison between zero-shot and finetune based TangoFlux inference on one text prompt.

affect the extent to which general TTA knowledge can be leveraged. These findings highlight our benchmark's unique role in uncovering such model-specific behaviors and point to an open research direction: how to best exploit general TTA knowledge to scale up relation-aware TTA, and conversely, how relation aware training can reciprocally improve general TTA. We visualize the generated audio

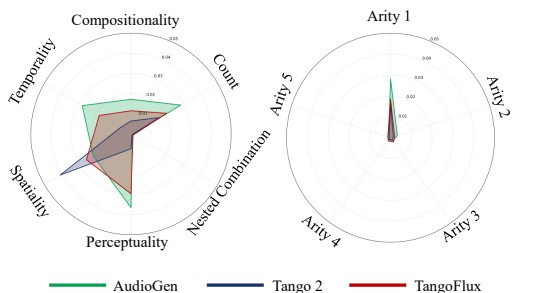
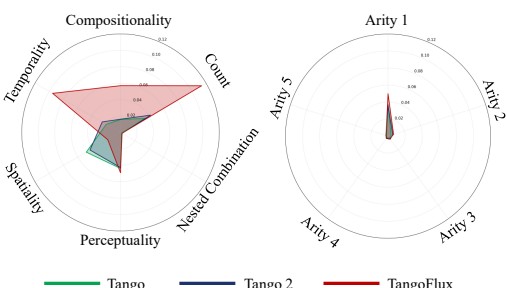

Figure 6: mAMSR regarding 6 main relation category and 5 relation Arity in Zero-shot setting.

Figure 7: mAMSR regarding 6 main relation category and 5 relation Arity in finetuning setting.

comparison between TangoFlux in zero-shot and fine-tuned inference modes in Fig. 5; this figure clearly shows that fine-tuning on our curated dataset benefits relation aware modeling.

To further investigate the role of training data size, we extend both finetuning and training-from-scratch experiments to larger datasets of 200 hours and 300 hours. As shown in Fig. 8, the mAMSR trend reveals two distinct behaviors: finetuning yields strong early gains but quickly saturates as the data size approaches 300 hours, whereas training from scratch continues to improve substantially with increasing data. This divergence underscores an important insight: scaling relation aware TTA models ultimately requires massive datasets, and reliance on finetuning alone may be insufficient for long-term progress. Our benchmark is therefore essential: it not only provides the controlled scaling environment needed to expose these trends, but also offers the first practical platform to systematically study how training strategy and data size interact in advancing relation aware TTA.

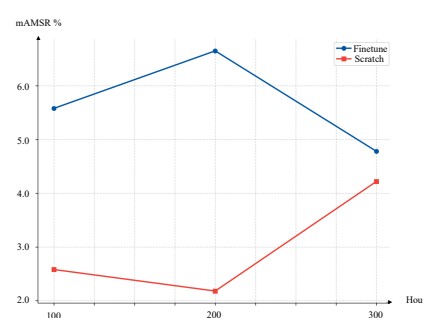

Figure 8: mAMSR variation w.r.t. training data size (100 h, 200 h and 300 h).

We visualize mAMSR across 5 main relation categories and relation arities for three strong methods in the zero-shot setting (Fig. 6) and the finetuning setting (Fig. 7). From these two figures, we observe that finetuning generally improves relation aware modeling capability. In the zero-shot setting, AudioGen (Kreuk et al., 2023) performs better than the other two on main categories including *Temporality*, *Count*, and *Perceptuality*. In the finetuning setting, TangoFlux (Hung et al., 2026) becomes the best-performing method. However, all methods in both settings perform poorly on *Nested Combination* or arity greater than 1. Our proposed benchmark enables researchers to tackle these challenges at scale.

## 4.6    MORE INVESTIGATION INTO EXISTING TTA MODELS

Relation aware TTA demands not only the correct presence of target audio events but also the faithful preservation of their underlying relations. However, current TTA methods (Hung et al., 2026; Ghosal et al., 2023; Xue et al., 2024) remain narrowly focused on single-event generation, leaving them ill-equipped to handle multi-event, relation-aware prompts. Table 4 makes this gap explicit: while TangoFlux (Hung et al., 2026), the state-of-the-art general TTA model,

Table 4: Audio event and relation accuracy of TangoFlux generation under different setting.

| Description | Accu. |
|---|---|
| Event (single event, no relation) | 75% |
| Event ( multi-event, relation aware) | 12% |
| Relation (multi-event, relation aware) | 3% |

achieves 75% accuracy on single-event prompts, its performance collapses to just 12% for multi-event correctness and a mere 3% for relation fidelity. This dramatic degradation exposes a fundamental blind spot in existing approaches: relation aware modeling is virtually unaddressed. Our benchmark directly targets this deficiency, offering the first systematic platform to quantify and dissect these failures. By doing so, it not only diagnoses the shortcomings of current TTA methods but also establishes the essential foundation for driving genuine advances in relation-aware TTA.

## 5 CONCLUSION AND DISCUSSION

This work presents *Aurelius* as a benchmark-centric foundation for advancing relation aware text-to-audio (TTA) generation. Across extensive zero-shot and adapted evaluations, we show that existing general-purpose TTA models remain fundamentally weak at relation modeling: while they can often produce plausible single events, performance drops sharply when prompts require multi-event composition, relational consistency, and higher-arity reasoning. This gap confirms that relation fidelity is not a minor extension of general TTA quality, but a distinct capability that current methods do not reliably learn.

To address this bottleneck, Aurelius contributes two complementary resources: *AudioEventSet* (110 clean, diverse, hierarchically structured events) and *AudioRelSet* (100 relations with explicit arity and compositional structure): together with a scalable `<text,audio>` pair construction pipeline and unified evaluation protocol. The benchmark enables controlled analysis across relation type, complexity, training strategy, and data scale, and therefore supports both diagnosis and method development.

We expect Aurelius to serve as a common testbed for relation aware generation, structured prompting, agentic decomposition, and new architectures that explicitly model event-relation dependencies. We also anticipate broader reuse in adjacent tasks within and beyond audio based machine learning. For example, *AudioEventSet* can be potentially used for adjacent acoustic tasks such as acoustic scene understanding, sound event detection and localization (He et al., 2021) and spatial audio modeling (He et al., 2024); *AudioRelSet* can benefit relation modeling in other research comunities including computer vision, natural language processing and multimodality. In short, Aurelius establishes a rigorous, extensible starting point for the next stage of structured TTA research, with the potential to motivate and advance other relevant research questions.

## 6 ACKNOWLEDGEMENT

We sincerely thank the reviewers for their constructive feedback and thoughtful engagement with this work, particularly for encouraging detailed clarifications during the multi-round discussion. Their input is invaluable for improving both the quality and clarity of this work.

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

# A APPENDIX

## .1 AUDIOEVENTSET DETAIL

The detail of *AudioEventSet* is given in Table I.

## .2 AUDIORELSET DETAIL

The detail of *AudioRelSet* is given in Table II.

## .3 AGENTIC WORKFLOW

The agentic workflow operates in three stages:

**1. Planning** Given the input text and the audio length to be generated, we query LLM to convert the input text into a JSON-formatted plan. The prompt is provided below:

---

### Planner Prompt

You are an audio scene planner for relation aware text-to-audio (TTA) generation. Perform all reasoning INTERNALLY and output ONLY valid JSON with the final scheduling results. Never include explanations or chain-of-thought. Your JSON MUST be parsable and follow the schema exactly.

OBJECTIVE

Given (1) a natural-language text prompt describing an audio scene with multiple events and relations, and (2) a target total duration in seconds, decompose the scene into concise sub-prompts (each corresponds to one independent audio event to be synthesized by a TTA model), and schedule them on the global timeline with start times and durations.

OUTPUT (JSON ONLY)

```
{
  "total_duration_sec": <float>,
// equals the requested total duration
  "sub_prompts": [
    {
      "id": "E1",                                // short unique id
      "text": "<short English sub-prompt>",
// head + present participle; may include light modifiers
      "start_sec": <float>,                      // >= 0
      "duration_sec": <float>
// > 0, and start_sec + duration_sec <= total_duration_sec
    }
    // ... typically <= items total
  ]
}
```

LANGUAGE FOR SUB-PROMPTS
- Use concise English in "head + present participle" style: e.g., "door bell ringing audio", "footsteps running audio".
- Add light, meaningful modifiers when clearly implied by text: "as background", "approaching", "departing", "slight reverberation", "time-stretched", "amplified", "attenuated", "balanced against X", "blended with Y".
- Avoid redundant words; keep each sub-prompt single-sentence and $\leqslant$ 18 words.

RELATIONS AND HOW TO REFLECT THEM (IMPLICITLY VIA SCHEDULING + WORDING)
- **Sequencing:** "then/after/next/first...then..." → schedule sequentially with a small gap $\approx$ 0.1s.

---

- **Simultaneity / Mix / Background** → allow overlaps; backgrounds can span large portions of the timeline.
- **Approaching / Departing** → keep in wording ("approaching"/"departing"). No mandatory duration change.
- **OR / XOR** → choose the most natural option; do not include the unchosen one.
- **NOT / prohibit** → exclude that event entirely.
- **IF...THEN...ELSE** → choose the most sensible branch; output only the chosen branch.
- **Count** → if an explicit number of items/events is requested, match it.
- **Repetition** → instantiate repeated events (e.g., bell ringing three times) as multiple sub-segments or one sustained segment if implied.
- **Proximity / Closeness / Farness** → reflect via wording only ("distant thunder", "near crowd"). No strict timing rules.

DURATION AND SCHEDULING RULES

- Respect total duration: sum of all segments should match the requested duration ($\pm0.25$s); if off, adjust proportionally.
- Each segment `duration_sec` MUST be integer.
- Choose reasonable segment durations:
    - transient cues (e.g., bell, door knock, gunshot) → 1–2s
    - medium actions (e.g., footsteps, typing, sawing) → 2–6s
    - ambient backgrounds (e.g., rain, wind, crowd murmur) → long spans (often entire duration)
- `start_sec` $\geqslant 0$; end $\leqslant$ total duration.
- On conflicts, preserve explicit relations first; compress lightly but keep segments $\geqslant$ 0.5s.
- Keep the number of sub-prompts concise (typically $\leqslant 5$).

FORMATTING RULES

- Output VALID JSON only. No comments, no trailing commas, no text outside JSON.
- Floats may be given with 1–2 decimals.
- Ensure "`start_sec` + `duration_sec` $\leqslant$ `total_duration_sec`" for all segments.

**2. Segment Synthesis**. Each sub-prompt is independently synthesized into audio by using TangoFlux (Hung et al., 2026).

**3. Stitching**. All generated audio segments are linearly composited by respecting their designated started time to generate the final audio.

## .4 TTA MODEL BASELINES INFERENCE SETTING

Table I: *AudioEventSet* corpus detail. We list all 110 event classes, which are deriving from 7 main categories and 23 sub-categories.

| Main Category | Sub-Category | Names | Description |
|---|---|---|---|
| Animal (22) | wild ground animal | lion roaring, wolf howling, donkey braying, cricket chirping, frog croaking, horse neighing | live in the wild |
| | domestic animal | dog barking, cat meowing, dog growling, cat spurring | live in domestic setting |
| | livestock | pig oinking, sheep bleating, cow mooing, rooster crowing, duck quacking | domesticated livestock |
| | wild animal | cuckoo calling, birds chorus, seagull cawing, peacock rattling blue jay whistling, nightingale singing, fly buzzing | animals in the wild |
| Human (21) | human voice | baby crying, laughing, shouting, whistling, coughing, snoring, sneezing, chewing, burping, farting | human use vocal tract |
| | human speech | male speech, female speech, child speech, group talk | speech audio |
| | hands action | finger snapping, clapping | audio by action |
| | group action | group clapping, cheering, group talking | audio by a group |
| | locomotion | running, footsteps | audio by movement |
| Machinery (13) | alarm | siren, door bell, car horn, bicycle bell, telephone ringing, telephone dialing, boat horn | machinery alarming |
| | mechanism | ratchet and pawl clicking, camera shuttering, printer printing, engine revving, clock ticking, paper shredding | mechanism audio |
| Human-Obj Interaction (18) | tools | hammer nailing, wood sawing, pen writing, wood chopping, rasping | human use tools |
| | culinary | dish audio, silverware audio, food frying, vegetable chopping | in kitchen setting |
| | work | toilet flushing, pouring water, keyboard typing, door slamming, cupboard open or close, drawer open or close, packing tape, dentist drilling, door knocking | audio during work |
| Obj-Obj Interaction (15) | impact audio | key jingling, ball bouncing, pen clicking, wind chime | impact effect |
| | friction audio | car emergency braking knife sharpening, sandpaper scraping, plastic scratching, string rubbing | friction effect |
| | dropping audio | coin dropping, glass clinking, metal dropping | dropping effect |
| | explosion | gunshot, firework, artillery fire | explosion effect |
| Music (11) | music instrument | plucked string, piano keyboard, bowed string, wind string, brass, harmonica, accordion | musical instruments |
| | singing | female singing, male singing, child singing, group singing | singing audio |
| Nature (10) | water | water bubbling, ocean wave, water dripping, water flowing, water boiling | water movement |
| | weather | thunder, wind, rain | nature weather |
| | nature change | wood cracking, rustling leaves | natural change |

Table II: *AudioRelSet* corpus detail. We introduce 21 basic relations, and advanced 79 nested combination relations, resulting in a total of 100 relations – 9 times larger than the relation corpus proposed in RiTTA (He et al., 2025). *AudioRelSet* maximumly covers all potential relations that audio events may exhibit in either the physical world or linguistic description. It is worth noting that *AudioRelSet* is open-ended. By nesting existing relations, we can potentially construct massive new relations.

| Category | Relation Name | Explanation | Event Arity | Sample prompt |
|---|---|---|---|---|
| Temporality (4) | precedence | before | binary | audio {A} followed by {B} |
| | succession | after | binary | create audio {A} after {B} |
| | simultaneity | same time | binary | {A} and {B} simultaneously |
| | periodicity | cyclic | unary | create audio {A} periodically |
| Spatiality (5) | closeness | spatial close | binary | {A} is closer than {B} |
| | farness | spatial far | binary | {A} is farther than audio {B} |
| | proximity | equal-dist | binary | {A} and {B} the same dist |
| | approaching | moving close | unary | {A} is moving closer |
| | departuring | moving away | unary | {A} is moving further away. |
| Count (1) | count | number | $n$-ary | 3 audios: {A}, {B} and {C} |
| Perceptuality (6) | balancing | level balance | binary | {A} dominates, {B} fades |
| | blending | mix audios | binary | {A} and {B} are mixed |
| | reverberation | reverberant | unary | generate audio {A} in canyon |
| | time-stretching | speed manipulate | unary | stretch audio {A} in time scale |
| | amplification | become louder | unary | amplify {A} to be louder |
| | attenuation | less loudly | unary | attenuate {A} to be quieter |
| Composition-ality (5) | conjunction | logical AND | binary | create both {A} and {B} |
| | disjunction | logical OR | binary | create {A} or {B}, or both |
| | negation | logical NOT | unary | do not generate audio {A} |
| | exclusive-or | logical XOR | binary | generate {A} or {B}, not both |
| | implication | if-then-else | ternary | if {A}, then {B}, else just {C} |
| Nested Combination (79) | Temp + Spat (4) | Temp + Spat | binary | {A} before approaching {B} |
| | Temp + Percep (8) | Temp + Percep | | reverb. {A}, succeeded by {B} |
| | Percep + Comp (12) | Percep + Comp | | stretched {A} or {B}, not both |
| | Spat + Comp (4) | Spat + Comp | | approaching {A} or {B}, not both |
| | Temp + Comp (6) | Temp + Comp | ternary | {A} first, then {B} or {C} |
| | Percep + Comp (1) | Percep + Comp | | mix {A} with {B}, or {C} |
| | Comp + Comp (1) | Comp + Comp | | {A} and {B}, or {A} and {C} |
| | Spat + Comp (5) | Comp + Comp | | {A} and {B}, or {A} and {C} |
| | Spat + Comp + Percep (2) | Comp + Comp | | {A} and {B}, or {A} and {C} |
| | Temp + Comp (4) | Temp + Comp | quaternary | audio {A} or {B} first, followed by {C} or {D} |
| | Comp + Comp (7) | Comp + Comp | | {A} or {B} first, then {C} or {D} |
| | Temp + Comp (3) | Temp + Comp | quinary | {A} before {B} first, then {C} before {D} or {E} |
| | Spat + Comp (9) | Spat + Comp | | if {A} closer than {B}, then {C} closer than {D}, else {E} |
| | Comp + Comp (9) | Comp + Comp | | if {A} and {B}, then {C} and {D} else {E} |
| | Count + Comp (4) | Count + Comp | | if {A}, {B}, {C}, then {D}, else {E} |

| Methods | Setting |
|---|---|
| AudioLDM (S-Full) (2023) | guidance_scale=5, random_seed=42, n_candidates=3 |
| AudioLDM (L-Full) (2023) | guidance_scale=5, random_seed=42, n_candidates=3 |
| AudioLDM 2 (L-Full) (2023) | guidance_scale=3.5, random_seed=45, n_candidates=3 |
| MakeAnAudio (2023) | ddim_steps = 100, scale = 3.0 |
| AudioGen (2023) | model name: audiogen-medium |
| Auffusion (2024) | num_steps = 100, guidance=7.5, num_samples=1 |
| LAFMA (2024) | num_steps = 200, guidance=3, num_samples=1 |
| Tango (2023) | num_steps = 200, guidance=3, num_samples=1 |
| Tango 2 (2024) | num_steps = 200, guidance=3, num_samples=1 |
| TangoFlux (2026) | num_steps = 50, guidance=3, num_samples=1 |

Table III: Detail setting for each TTA method.

