# OpenReview forum: "Aurelius: Relation Aware Text-to-Audio Generation At Scale"
_ICLR.cc/2026/Conference — ICLR 2026 Poster_

### Official Review · Reviewer_P1t4 · 2025-10-16

**Soundness:** 3
**Presentation:** 2
**Contribution:** 2
**Rating:** 4
**Confidence:** 4

**Summary:**

This paper proposes a relation-aware text-to-audio generation framework and introduces two newly developed datasets, AudioEventSet and AudioRelSet, to improve the modeling of relational structures between audio events. The datasets are organized in a tree-structured hierarchy and are designed to enhance compositionality and relation understanding in audio generation. The authors argue that these datasets can serve as new benchmarks for relation-aware audio generation.

**Strengths:**

The paper focuses on an important and relatively underexplored topic — relation-aware text-to-audio generation, which aims to capture richer dependencies between sound events beyond conventional text-to-audio mapping.

The proposed datasets include explicit relation-level annotations (such as “arity,” count, and compositional structure), which provide more fine-grained relational information than most existing datasets.

The authors attempt to model hierarchical relations between sound events, potentially offering insights into how complex auditory scenes could be represented in structured data formats.

**Weaknesses:**

The main contribution lies almost entirely in dataset creation, with no substantial methodological innovation or new generation framework beyond existing relation-aware approaches (e.g., RiTTA or CompA).

The datasets are not released, and only high-level descriptions are given. Without access to examples or samples, it is impossible to verify the data quality or reproducibility.

The tree structure design (depth 3 for AudioEventSet and depth 2 for AudioRelSet) lacks theoretical or empirical justification. It is unclear why this specific hierarchy benefits the relation-aware task or how the root/leaf nodes are defined.

The data generation process appears mostly manual or GPT-assisted, rather than automatic. This makes it resource-intensive, potentially inconsistent, and difficult to scale. The resulting compositions may sound unnatural or ambiguous, especially since all clips are fixed to 10 seconds regardless of content or event duration.

The evaluation methodology is largely inherited from general text-to-audio works and RiTTA, without introducing new metrics to evaluate the proposed hierarchical or relation-specific structures.

Overall, the paper reads more like an engineering dataset report than a research paper introducing conceptual or algorithmic innovation.

**Questions:**

Previous works such as RiTTA and CompA also address relation-aware text-to-audio generation. Beyond the newly developed datasets, what is the core conceptual or technical difference between this work and prior ones?

How exactly were the datasets collected — are all audio-text pairs manually constructed or verified? If so, how was data distinctiveness and quality control ensured?

Did the authors listen to and validate all audio compositions to ensure naturalness and correct temporal ordering of events?

The proposed tree structure seems to play no role in evaluation — could the authors propose or experiment with metrics that explicitly utilize this structure?

Why was the relation-aware improvement only tested on the Tango model family? Would results generalize to other text-to-audio architectures (e.g., AudioLDM, Make-An-Audio)?

Is there any other way to improve the performance on relation aware?  Not just training on the "new-proposed dataset".

Any demo or public data for the dataset (waveform)?

---

> ### Author Response · Authors · 2025-11-22
> **Author Feedback**
>
> We thank the reviewer for the constructive review. We provide detailed feedback below to address your concerns.
>
> 1. **Q1:** The dataset is not released.
>
>    **A1:** As we presented in the main paper, we will release all AudioRelSet, AudioEventSet and text-audio pair generation pipeline to the public to facilitate relation-aware text-to-audio generation research.
>
> 2. **Q2:** what is the core conceptual or technical difference between this work and prior ones.
>
>    **A2:** we submit this work to ICLR **datasets and benchmarks** track, which means the main focus in the paper is to introduce a novel benchmark that can be used to systematically investigate relation-aware text-to-audio generation task. The technical contribution goes beyond the scope of this work. Based on our study and also agreed by RiTTA paper, the introduction of large-scale benchmark is the key issue for relation-aware text-to-audio generation research at scale, we thus aim to fill in this gap in this work.
>
> 3. **Q3:** How were the datasets collected? Were all audio–text pairs manually constructed or verified? How was quality/distinctiveness ensured?
>
>    **A3:** AudioEventSet and AudioRelSet were curated through a hybrid manual + programmatic process:
>
>    For **AudioEventSet**:
>
>    * All raw audio candidates were sourced from copyright-free Freesound or FSD50K.
>
>    * Each event class has ~75 manually verified clips.
>
>    * Verification ensures correct labeling; single-event (non-polyphonic) audio; no ambiguous or overlapping sources; consistent acoustic identity within a class; Classes that are inherently ambiguous (e.g., engine idling or noisy crowd events) were removed.
>
>    For AudioRelSet:
>
>    * Relation definitions are hand-designed, and each relation has 5–6 manually curated or GPT-assisted templates.
>    * Each template is manually reviewed to ensure semantic correctness, linguistic clarity, and faithful expression of the relation.
>
>    Thus, both corpora undergo strict human verification steps that ensure distinctiveness and high quality, **we have spent about one year to curate the benchmark**.
>
> 4. **Q4:** Did the authors listen to and validate all audio compositions to ensure naturalness and correct temporal ordering?
>
>    **A4:** Yes. During validation:
>    * We sampled a substantial number of synthesized <text,audio> pairs across all relation types, especially temporality and nested relations.
>    * For each sampled pair, we manually listened and checked:
>    1. whether the correct events appear;
>    2. whether ordering (before/after/overlap) is respected;
>    3. whether mixing, reverb, time-stretching, or amplification is perceptually coherent.
>
>    We also validated the audio event detector and acoustic-effect classifier (used in evaluation) on large manually inspected sets to ensure reliable relation extraction.
>
> 5. **Q5:** The tree structure seems unused in evaluation — can metrics explicitly utilize it?
>
>    **A5:** You correct that the current evaluation does not explicitly leverage the tree. The hierarchy primarily serves data construction, class separation, and relation scalability. In the experiment, we just report the result across all leaf nodes due to the space limit. It is definitely feasible to report more results across w.r.t different tree structure (hierarchies). We will add more structured report in the revised version.
>
> 6. **Q6:**: Why was relation-aware improvement only tested on the Tango family? Would results generalize to AudioLDM/Make-An-Audio/etc.?
>
>    **A6:** We chose Tango/TangoFlux/Tango2 because:
>    * they represent state-of-the-art text-to-audo generation models up to date,
>    * they achieve the best zero-shot performance in relation-aware metrics (Table 2),
>    * we experimentally find fine-tuning and training from scratch on Tango family give better result than other methods like AudioLDM/Make-An-Audio.
>
>    Therefore, we report Tango Family result in the main paper and treat it a strong baseline for future development.
>
> 7. **Q7:** Is there any way to improve relation-aware performance besides training on the new dataset?
>
>    **A7:** Yes. We outline several alternatives, some are discussed in Sec. 4.6. We have been working on and testing them after this work submission.
>
>    * Modular generation: separate event generation and relation conditioning and combine them together at a later stage.
>    * Architecture-level changes: adding relation token and audio event tokens, further incorporate attention to merge them together to output the final relation-aware audio.
>
>    Our introduced Aurelius benchmark is only the first step; the benchmark is specifically designed to motivate new architectures and training strategies.

---

> > ### Comment · Reviewer_P1t4 · 2025-11-22
> >
> > Thank you for your response. Overall, the task still relies heavily on manual effort and does not introduce substantial novelty. However, since this submission is in the dataset track, and it is clear that the authors have invested significant time and effort into building the dataset, I have adjusted my score upward. Good luck with your submission.

---

> > > ### Author Response · Authors · 2025-11-23
> > > **Thanks for reviewing our feedback**
> > >
> > > Dear Reviewer,
> > >
> > > Thanks for your kind follow-up review of our feedback, and recognizing our effort in curating this benchmark. We wish you all the best with your research.
> > >
> > > Thanks,
> > > Authors

---

### Official Review · Reviewer_2P2L · 2025-10-30

**Soundness:** 3
**Presentation:** 3
**Contribution:** 3
**Rating:** 4
**Confidence:** 4

**Summary:**

This paper introduces Aurelius, a new framework designed to enable relation-aware text-to-audio (TTA) generation research at scale. The authors also propose two accompanying datasets, AudioRealSet and AudioEventSet, which demonstrate effectiveness in audio generation tasks. Overall, the proposed datasets are valuable contributions to the audio research community. However, the paper lacks novel methodological and technical innovations beyond dataset construction. (But the dataset is a great one!)

**Strengths:**

- The release of a large-scale, well-curated dataset is always beneficial to the audio research community.

- The authors conduct comprehensive benchmarking of existing TTA models to evaluate the proposed datasets.

- The distinction between AudioEventSet and AudioRealSet is clearly defined. In particular, AudioRealSet provides rich attribute-level annotations for sound events, which is a valuable addition.

**Weaknesses:**

- My main concern is that, despite the usefulness of the dataset, there are no novel methodological or technical contributions. The paper reads primarily as a dataset paper rather than a technical paper.

- In the Introduction, the term “relation modeling” is not clearly defined.

- In Section 3.1, the phrase “audio events potentially present in the 3D physical world is unclear” is ambiguous — clarification is needed.

- The authors claim that AudioEventSet is more distinctive than AudioSet, but no supporting evidence or analysis is provided. Since AudioEventSet is also manually designed, how do the authors ensure that the 110 sound classes are well-separated and not confusing?

- The size of AudioEventSet should be reported more clearly. As it is built from Freesound and FSD50K (which contains only ~50k clips), it is unclear how large the final dataset actually is or how scalability is achieved.

- In Section 3.2, is AudioRealSet a subset of AudioEventSet? How are the relations labeled — automatically or manually? The writing in this section could be improved for clarity.

- The experimental section lacks comprehensive evaluation of how the proposed datasets improve performance. For instance, how much improvement is observed when models are trained with these datasets compared to without them?

**Questions:**

N.A.

---

> ### Author Response · Authors · 2025-11-22
> **Author Feedback**
>
> We thank the reviewer for the constructive and thoughtful feedback for our work. We address the main concerns raised by the reviewer in below:
>
> 1. **Q1:**: The paper reads primarily as a dataset paper, with no novel methodological or technical contributions.
>
>    **A1:** We submit this work in **datasets and benchmarks** track, which aligns well with the aim of this track. Furthermore, we have provided detailed technical contributions in the whole benchmark curation, ensuring the diversity, high-quality and large-scale characteristic of our benchmark.
>
> 2. **Q2:** In the Introduction, ‘relation modeling’ is not clearly defined.
>
>    **A2:** We will revise the Introduction to explicitly define relation modeling as:
>    “The process of interpreting and generating audio that satisfies the structural relations—temporal, spatial, logical, compositional, perceptual, or nested—specified by the text.”
>
>    We also point the reviewer to Sec. 3.2, which introduces the concrete structure of relation modeling (Temporality, Spatiality, Count, Perceptuality, Compositionality, Nested Combination) and describes how relations influence audio generation.
>
> 3. **Q3:** In Section 3.1, ‘audio events potentially present in the 3D physical world’ is unclear.
>
>    **A3:** We thank the reviewer for the suggestion. The intended meaning is:
>
>    “We aim to cover the types of real-world audio events that humans commonly encounter in everyday 3D physical environments, such as humans, animals, weather, tools, machinery, object interactions, and musical sources.”
>
>   This is clarified by the ontology structure in Fig. 2A and the full hierarchy in Table I (appendix), which explicitly enumerates the 110 events organized by physical origin (e.g., water flow, object impacts, mechanical operations, human speech, animal calls). We will rewrite the sentence to remove ambiguity.
>
> 4. **Q4:** Authors claim that AudioEventSet is more distinctive than AudioSet, but no supporting evidence is given. How do the authors ensure the 110 classes are well-separated?
>
>    **A4:** AudioEventSet achieves distinctiveness through hard constraints in the curation protocol:
>
>    1. Ambiguous or overlapping classes are removed (e.g., engine idling, which overlaps with hairdryer or fans; see Sec. 3.1).
>
>    2. Each candidate class undergoes manual verification to ensure:
>    * event-specific acoustic signature;
>    * minimal overlap with other classes,
>    * high intra-class consistency.
>
>    3. Tree-structured ontology (coarse → fine) guarantees separation:
>    * 7 main categories
>    * 23 subcategories
>    * 110 leaf nodes
>
>    4. Class-level acoustic distinctiveness is validated through:
>    * removal of multi-source or polyphonic samples,
>    * removal of ambiguous acoustically similar categories,
>    * strict intra-class homogeneity rules (Sec. 3.1).
>
> 5. **Q5:** The size of AudioEventSet should be reported more clearly.
>
>    **A5:** AudioEventSet consists of:
>    * 110 fine-grained sound classes
>    * ~75 curated audio snippets per class
>    * Total ≈ 8,000 high-quality waveforms
>
>    AudioEventSet is curated from Freesound (which contains hundreds of thousands of clips), FSD50K as a supplementary. We further introduce manual filtering + deduplication to ensure AudioEventSet quality. The scalability is achieved because the text-audio pair-generation pipeline decouples audio events and relation, the new audio events can be easily added to the pair-generation pipleline to scale up the text-audio pair datasets.
>
> 6 **Q6:** In Section 3.2, is AudioRelSet a subset of AudioEventSet? How are relations labeled? Writing unclear.
>
>    **A6:** We clarify:
>
>    1. **AudioRelSet is not a subset of AudioEventSet**:
>    * AudioEventSet = audio event ontology (110 classes)
>    * AudioRelSet = relation ontology (100 relations)
>    * They are independent but combinable (Fig. 2C, via arity-based instantiation);
>
>    2. **How relations are labeled?** Relations are not labeled from real audio. Instead:
>    * Relations are defined symbolically in AudioRelSet.
>    * Each relation has 5–6 hand-designed or GPT-assisted templates.
>    * Each template is instantiated using real event names + synonyms.
>    * Nested relations undergo algorithmic logic-validation before inclusion.
>
>    Thus, AudioRelSet is manually and programmatically curated, not auto-labeled from noisy sources.
>
> 7. **Q7:** Experiments do not clearly show how the dataset improves performance compared to without it.
>    **A7:** We directly demonstrate improvement:
>    1. Zero-shot vs. finetuning vs. scratch (Table 2, 3): Training on Aurelius improves relation-aware scores by large margins. e.g., TangoFlux mAPre: Zero-shot: 12.38 →; Finetune: 28.57 (↑ 131%) →Scratch: 16.68.
>    2. Scaling study (Fig. 6). Training from 100 h → 200 h → 300 h shows progressive improvements using the curated data.
>    3. Event vs. relation correctness gap (Table 4). The dataset exposes failure modes (event accuracy 75% vs. relation accuracy 3%) and enables systematic improvement.

---

> > ### Author Response · Authors · 2025-11-27
> > **Follow-up on the author feedback**
> >
> > Dear Reviewer  2P2L
> >
> > We thank you again for your efforts in reviewing our paper. We have provided a detailed feedback addressing your comments and concerns. We would truly appreciate it if you could kindly take a look and share any further thoughts when you have time, and update the review accordingly. We highly value your feedback, and any further suggestions or clarifications would greatly help us strengthen the work.
> >
> > Thank you for investing time and effort in reviewing our work.
> >
> > Authors

---

### Official Review · Reviewer_SfjC · 2025-11-01

**Soundness:** 3
**Presentation:** 3
**Contribution:** 2
**Rating:** 4
**Confidence:** 3

**Summary:**

The paper presents Aurelius, which contains:
(1) AudioEventSet, which contains clean clips for audio events
(2) AudioRelSet, which contains the relative relationship of audio events.

The authors show that, by combining these two resources, we can generate theoretically unlimited relation-aware text-audio pairs.

The authors also present the evaluation method (benchmarks) for the relation-aware audio.

It also shows that training existing TTA models on the proposed dataset can benefit the relation-aware TTA performance.

The relation-awareness is an important property in current TTA, and this paper is a very good resource.

**Strengths:**

The paper comprehensively discusses the relation-aware TTA generation. It contains good resources, benchmarks and sufficient discussion.

The relation-awareness is of wide interest in the TTA community. The paper also reveals that the current TTA is not good enough in this direction.

**Weaknesses:**

(1) The paper is mostly about the resources (data, benchmark) of building relation-ware TTA. For this kind of resource paper, I wonder if the author would make it public.
(2) In section 3, although the author provides a comprehensive design in the data content, they don't mention (1) why the designs are reasonable and (2) how they ensure the intended design philosophy is well implemented in practice. e.g., how they ensure the audio clips in AudioEventSet are precise and clean enough; why such designs in AudioRelSet are reasonable. Such missing information would compromise the contribution of the work.
(3) I'm a bit confused about the mAMSR metric: in the Table 2 caption, you mention its range is [0, 1], but numbers in Table 3 are beyond this range.
(4) Even with this carefully designed data pipeline, the overall mAPre, mARel, etc, are still absolutely low. (e.g., <30%). This very tailor-made dataset seems not to solve the issue very well.
(5) The authors claim that unlimited data simulation is feasible, but scaling up the simulated data is not very effective, as shown in Figure 6.

**Questions:**

In general, whether the resources would be made public is an important metric for the paper evaluation. Would the authors release them?

---

> ### Author Response · Authors · 2025-11-21
> **Author feedback**
>
> We sincerely thank the reviewer for the thoughtful and constructive feedback. We respond to each point in detail below.
>
> 1. **Q1:** The paper is mostly about resources. Will the authors make it public?
>
>    **A1:** Yes, we will publicly release AudioEventSet, AudioRelSet, and the full pair-generation pipeline. These resources are explicitly designed as a community benchmark for relation-aware TTA research. It is with our original motivation to release the whole dataset and code to facilitate the relation-aware text-to-audio generation task.
>
>    AudioEventSet is curated from copyright-free or license-permissive sources with manual verification (Sec. 3.1) and AudioRelSet consists entirely of author-curated text templates and relation definitions (Sec. 3.2). Both corpora are reproducible and fully release-ready.
>
> 2. **Q2:** Why are the designs reasonable? How do the authors ensure the intended philosophy is implemented?
>
>    **A2:** **why are the designs reasonable?** Sec. 3.1 identifies four essential properties of a relation-aware TTA audio event corpus: diverse, high-fidelity, distinctive, and hierarchically structured. Existing datasets fail to satisfy these simultaneously (Table 1), largely due to noise, polyphony, label-missing issues, or semantic ambiguity. Aurelius therefore provides:
>     * Clean, single-event, unambiguous clips
>
>    * A coarse-to-fine ontology (Fig. 2A)
>
>    * Diversity via 75 curated clips per event class
>
>    * Event distinctions necessary for relation modeling (e.g., spatial/temporal relations cannot be inferred from polyphonic audio)
>
>    These design principles directly support relation reasoning:
>
>    **Ensuring AudioEventSet quality in practice**. We guarantee correctness and cleanness through:
>    * Manual verification of every audio clip: label correctness; absence of overlapping events; high signal quality.
>    * Exclusion of ambiguous classes (e.g., engine idling vs hairdryer).
>
>    **Reasonableness and correctness of AudioRelSet**. AudioRelSet is built from six fundamental relation categories: Temporality, Spatiality, Count, Perceptuality, Compositionality, and Nested Combination (Fig. 3). These categories map directly to relations observed in the physical world and natural text.  We ensure correctness by:
>
>    * Explicit internal logic and feasibility checks during nested relation construction
>
>    * Rejecting contradictory or ill-defined combinations (Sec. 3.2)
>
>    * Carefully crafting text templates with placeholder-based instantiation
>
>    Thus the relation corpus faithfully reflects the intended design philosophy.
>
> 3. **Q3:** mAMSR is stated to be in [0,1], but numbers in Table 3 exceed 1.
>
>    **A3:** This is a notation mismatch, not a metric error. In the table, we wrote: **mAMSR (%)**. Hence, a value such as 2.73 means 2.73%, consistent with the range. We will clarify this clearly in the camera-ready.
>
> 4. **Q4:** Even with the designed pipeline, mAPre/mARel are low (<30%). So the dataset doesn’t solve the issue.
>
>    **A4:** **This is expected and is exactly why Aurelius is needed.** Aurelius is not designed to make existing models perform well; it is designed to expose their fundamental limitations in relation reasoning. For example: Zero-shot methods achieve <10% across all relation-aware metrics (Table 2); Even after finetuning, models still struggle (<30%). Table 4 shows: 75% accuracy for single-event, 12% for multi-event, 3% for relation correctness. These results confirm that current TTA models lack relation modeling capabilities, validating the need for Aurelius.
>
> 5. **Q5:** Scaling simulated data is not effective (Fig. 6), contradicting the "unlimited simulation" claim.
>
>    **A5:**  Our claim is about feasibility of unlimited simulation, not that any model will scale linearly without diminishing returns. Fig. 6 shows: Finetuning saturates early → limited by inherited inductive bias. Training from scratch improves significantly with more data
>
>    Thus, scaling is indeed effective when the architecture is appropriate. Finetuning alone is insufficient for long-term scaling. Aurelius provides the first controlled platform to reveal this divergence.
>
> If you need further clarification or discussion, please feel free to let us know, we are happy to provide additional details or explanation.

---

> > ### Author Response · Authors · 2025-11-27
> > **Follow-up on Author Response**
> >
> > Dear Reviewer SfjC,
> >
> > We thank you again for spending time reviewing our paper. We have provided a detailed feedback and information addressing your comments and concerns. We appreciate it if you could kindly take a look and share any further thoughts. We highly value your feedback, and any further suggestions or clarifications would greatly help us to continue to improve our work.
> >
> > Thank you for investing time and effort in reviewing our paper.
> >
> > Authors

---

### Official Review · Reviewer_hVjf · 2025-11-01

**Soundness:** 2
**Presentation:** 2
**Contribution:** 2
**Rating:** 4
**Confidence:** 2

**Summary:**

This paper introduces Aurelius, a large-scale benchmark framework with two corpora, AudioEventSet and AudioRelSet, that enables systematic evaluation and development of relation-aware text-to-audio generation at scale.

**Strengths:**

- The paper tackles the well-established limitation of current TTA models in generating audio with accurate temporal ordering and relational structures, which is an interesting and important research problem.
- The framework's approach of combining relation templates with audio events to generate numerous <text, audio> pairs provides excellent flexibility and scalability. The adoption of the "Head-Modifier Structure with Progressive Verb Form" (e.g., "door bell ringing audio" rather than "ringing door bell") ensures syntactic consistency across the dataset.

**Weaknesses:**

- The GPT-generated templates or synonyms are not always accurate, and some generated texts may not properly correspond to the actual audio events, leading to potential noise in the dataset.
- Since the generated sounds are synthetic, there is no clear way to assess or guarantee their perceptual quality.
- Compared to the existing datasets shown in Table 1, the improvement in performance is not clearly demonstrated. There is no head-to-head comparison between models trained on existing datasets and those trained on Aurelius, making it difficult to quantitatively verify the superiority of the proposed dataset.

**Questions:**

Please refer to the weaknesses mentioned above.

---

> ### Author Response · Authors · 2025-11-21
> **Feedback by Authors**
>
> We thank the reviewer for the constructive review and feedback for your work. We address your listed weaknesses in below.
>
> 1. **Q1:** Potential noise in GPT-generated templates or synonyms.
>
>    **A1**: As stated in L290–293, we either manually write the relation-aware text templates or use GPT as an assistant for template generation. Importantly, every GPT-generated template undergoes three-stage human cross-verification to ensure correctness and coherence. Although a small portion (less than 1/20) of GPT-generated prompts initially contained noise, we manually corrected them to maintain high quality.
>    In the supplementary material, we provide AudioRelSet.json, which includes the curated templates for all relations. You are welcome to inspect them to assess their quality.
>
> 2. **Q2**: No clear way to assess perceptual quality of synthetic audio.
>
>    **A2**: For each generated sound, we associate it with a real (authentic) reference sound. We evaluate perceptual quality using standard metrics such as FAD, KL, and FD, which statistically compare generated audio against reference audio. Lower scores on these metrics indicate that the generated sound is closer to the reference sound, and therefore of higher perceptual quality.
>
> 3. **Q3**: Lack of head-to-head comparison with models trained on existing datasets.
>
>    **A3**: The purpose of Table 1 is to highlight the impracticality of using existing datasets for large-scale relation-aware text-to-audio generation.
>
>    * Existing datasets (AudioSet, FSD50K, AudioCaps, AudioTime) are polyphonic, noisy, have missing labels, and contain semantic ambiguity, making them unsuitable for relation-aware generation.
>
>    * Moreover, these datasets were not designed for relation-aware TTA; their text prompts do not contain relation semantics, making a direct head-to-head comparison infeasible.
>
>    Thus, Table 1 serves to motivate our proposed dataset Aurelius, which is high-quality, distinctive, and diverse, and provides an appropriate testbed for developing and evaluating relation-aware text-to-audio generation models.
>
> If you need further clarification or discussion, please feel free to let us know—we are happy to provide additional details.

---

> ### Comment · Reviewer_hVjf · 2025-11-24
> **Response to Authors**
>
> The concern regarding Q1 has been resolved; however, the audio quality still cannot be fully assessed beyond the reported experimental tables. Many papers provide qualitative analysis, but in the audio domain, it is impossible to verify the actual quality without access to audio samples (not necessarily the entire set, but at least a few). In Q3, it remains unclear whether the claimed high-quality, distinctive, and diverse audio samples actually exist, especially compared to the original benchmarks.

---

> > ### Author Response · Authors · 2025-11-24
> > **Further Feedback by Authors**
> >
> > We thank the reviewer for the further reponse. Here we provide more details to address your concern.
> >
> > 1. **Audio Sample to Verify the Quality**.
> >
> >    **Feedback:** We appreciate the reviewer’s interest in validating the audio quality—this is indeed crucial for a benchmark intended to support the community. To facilitate direct inspection, we have uploaded **seed_audios.zip** in the updated rebuttal supplementary material. This package contains $\approx 170$ audio samples spanning a wide range of AudioEventSet classes.
> >
> >    You are welcome to download and listen to these samples to get a concrete sense of the audio fidelity, clarity, and distinctiveness. If you encounter any issues accessing the files or would like to examine additional samples from specific categories, please let us know—we are more than happy to provide further examples.
> >
> > 2. **it remains unclear whether the claimed high-quality, distinctive, and diverse audio samples actually exist, especially compared to the original benchmarks**
> >
> >    **Feedback:** We thank the reviewer for raising this important clarification. We provide more concrete evidence supporting the actual quality, distinctiveness, and diversity of AudioEventSet.
> >
> >    **First**, as highlighted in our earlier response, existing benchmarks such as AudioSet, AudioCaps, FSD50K, and AudioTime suffer from fundamental deficiencies: they are often polyphonic, noisy, missing accurate labels, or semantically ambiguous. These issues render them unsuitable for relation-aware TTA, which requires clean, single-source, unambiguous events. This motivates the design of AudioEventSet as a high-quality, diverse, and distinctive corpus.
> >
> >    **Second,** Second, the curation process of AudioEventSet was explicitly designed to address these deficiencies:
> >    * Each audio class in AudioEventSet is required to exhibit distinctive, clean, and cohesive acoustic characteristics. No two classes are allowed to sound acoustically similar, and no class is permitted to contain inconsistent or heterogeneous audio. For example, as noted in L157–L159, “engine idling” in AudioSet varies drastically across vehicle types and often overlaps acoustically with fans or hairdryers; we therefore exclude such ambiguous classes from our ontology.
> >    * We also aim for comprehensive coverage of commonly heard everyday sounds. To achieve this, we construct a hierarchical audio-event tree, gradually expanding it to include more fine-grained categories across humans, animals, nature, machinery, music, and interaction-based events.
> >    * In practice, we manually examined class names and categories from AudioSet and FSD50K, and supplemented them with additional classes proposed by us when gaps were identified. For each candidate class, we actively searched for multiple real-world audio exemplars from Freesound or FSD50K, and manually verified that they meet our criteria of being high-quality, distinctive, and unambiguous. Any class failing these checks was discarded.
> >
> >    The entire curation process was meticulous and time-intensive, spanning roughly one year. Throughout this period, we repeatedly revisited and refined the ontology, evaluating candidate nodes and determining whether to keep or remove them based on the strict acoustic criteria above. This extensive manual effort ensures that 110-class AudioEventSet is genuinely high-quality, well-separated, and suitable for relation-aware generation.
> >
> > We are happy to answer more of your concerns or provide more details, feel free to let us know if there is any.

---

> > > ### Comment · Reviewer_hVjf · 2025-11-27
> > > **Response to the authors**
> > >
> > > I appreciate the authors for providing the audio samples, which appear very clear and diverse as seed audios. I believe that one of the novelty claims of this paper lies in Section 3.3, Text–Audio Pair Creation, where the seed audios serve as sources for TTA audio generation (If it is not, please feel free to correct me). Given the instantiated text and the corresponding seed audio, if the authors provide the generated audio–text pairs, it will be greatly helpful in updating my score.

---

> > > > ### Author Response · Authors · 2025-11-27
> > > > **Author feedback to text-audio pairs dataset**
> > > >
> > > > We thank the reviewer for confiming the high-quality and diversity of our curated seed audios.  Yes, we use the curated seed audio to further generate the text-audio pair data, but the generation pipeline maximumly ensures the text and audio diversity in the generated text-audio pairs.
> > > >
> > > > Following your advice, we updated the supplementary material and added the audio-text samples. You can find the folder **textaudio_pairs**, within which you can find **textaudio_pair.json** that records the text prompt and its corresponding reference audio name saved in **audios** subfolder. We provide about 200 audio-text pairs for your reference, note that some text prompt may correspond to multiple reference audios but we just provide one sample audio here for simplicitly.
> > > >
> > > > If you need more detail regarding this benchmark, or have any concern while experiencing the audio-text pairs, feel free to let us know.

---

### Author Response · Authors · 2025-11-23
**General feedback by authors**

We sincerely thanks all reviewers for investing time and effort in reviewing our paper, and further providing with constructive feedback. We here provide a general feedback to address the common concerns.

1. **Technical Contribution Concern**

   **Feedback:** We appreciate the reviewer’s perspective and fully agree that developing new methods for relation-aware TTA is an exciting direction. However, our submission is targeted specifically to the ICLR **Datasets and Benchmarks** track, where the primary evaluation criteria focus on the quality, scalability, necessity, and impact of the proposed benchmark rather than on novel architectural advances.

   Within this scope, our work (Aurelius) makes several susbstantive contributions, including a large-scale audio event corpus AudioEventSet, large-scale relation corpus AudioRelSet, and a scalable, reproducible <text,audio> pair generation engine, and extensive evaluation to date of existing TTA models under relation-aware setting. These contributions are central to the "Dataset and Benchmarks" track.

   We agree that new technical methods built upon Aurelius would further advance the field; indeed, we are actively exploring such architectures. But these developments are orthogonal to the core purpose of this paper—which is to define, standardize, and enable research on relation-aware TTA at scale. The absence of a novel method does not diminish the value or necessity of the benchmark, and is fully aligned with the expectations of this track.


2. **Difference from other benchmarks like AudioSet.**

   **Feedback:**  We appreciate the reviewer’s question regarding how Aurelius differs from existing audio datasets. As highlighted in Table 1 of the main paper, the widely used datasets (AudioSet, FSD50K, AudioCaps, AudioTime) are not suitable for relation-aware TTA due to several fundamental issues:

   * **Polyphony and noise**: Many clips contain multiple overlapping sources, environmental noise, or uncontrolled recording conditions, making it impossible to extract clean, single-event audio suitable for modeling fine-grained relations.
   * **Missing or Unreliable Labels**: AudioSet in particular has weak, human-in-the-loop labels that often omit events or mislabel polyphonic mixtures.
   * **Semantic ambiguity**: Some class labels inherently lack consistent acoustic identity. As stated in L156–158, “engine idling in AudioSet differs significantly by engine type and can easily be confused with a working fan or hairdryer.” This ambiguity makes relational modeling (e.g., temporal order, proximity, compositionality) infeasible.
   * **Lack of hierarchical and relation-aware design**: None of the existing datasets provide a clean event ontology, arity-aware relation labels, or text templates capable of generating structured relational descriptions.

   Although AudioSet appears large in class count, a substantial portion of its categories is too acoustically inconsistent or noisy to serve as reliable building blocks for relation-aware generation. In contrast, **AudioEventSet** was curated explicitly to provide clean, distinctive, non-polyphonic, and human-recognizable events—each manually validated for correctness and separability. Therefore, existing datasets cannot replace AudioEventSet for relation-aware TTA at scale. **It took us about one whole year to curate the AudioEventSet and AudioRelSet**.

3. **Will the benchmark be released?**

   **Feedback:** Yes. As stated in the main paper, we will publicly release AudioEventSet, AudioRelSet, and the complete text–audio pair generation engine code. Our goal is to establish Aurelius as a standard benchmark for relation-aware TTA.

   To further support the community, we will additionally:
   * maintain a public leaderboard for tracking model performance on relation-aware metrics;
   * release all code and metadata required for full reproducibility of <text,audio> pair generation;
   * provide clear documentation to support extension of the ontology with new events or relations.

   We believe that an open, extensible benchmark is essential for catalyzing research in structured audio generation and cross-modal relational reasoning.

---

### Author Response · Authors · 2025-12-02
**Post-rebuttal summary for Area Chair**

Dear Area Chair,

In light of ICLR26 freezing reviewer update, we would like briefly summarize our rebuttal in addressing the reviewers' main concerns. During the author-reviewer discussion period, **reviewer #P1t4 has raised the score the 6**, and **reviewer #hVjf has clearly expressed willingness to inscrease the score** after checking the quality of the seed audios in the benchmark (the reviewer update has been disenabled right after we provide more details). For #SfjC and #2P2L, they didn't repond or adjust the score to our rebuttal, although we tried best and addressed their concerns long before the discussion closes.  Here we address the common concerns raised by  #SfjC and #2P2L, as well as by the other two reviewers.

1. **Will the dataset be released?** raised by #SfjC and #P1t4.

   **Answer:**   Yes. As stated in the main paper, we will publicly release AudioEventSet, AudioRelSet, and the complete text–audio pair generation engine code. Our goal is to establish Aurelius as a standard benchmark for relation-aware TTA.

   To further support the community, we will additionally:

   * maintain a public leaderboard for tracking model performance on relation-aware metrics;
   * release all code and metadata required for full reproducibility of <text,audio> pair generation;
   * provide clear documentation to support extension of the ontology with new events or relations.

   We believe that an open, extensible benchmark is essential for catalyzing research in structured audio generation and cross-modal relational reasoning.

2. **Technical Contribution**, raised by #2P2L and #P1t4.

   **Answer:**  Aurelius is submitted to the ICLR Datasets and Benchmarks track, where the primary evaluation criteria focus on the quality, scalability, necessity, and impact of the proposed benchmark rather than on novel architectural advances.

   Within this scope, our work (Aurelius) makes several susbstantive contributions, including a large-scale audio event corpus AudioEventSet, large-scale relation corpus AudioRelSet, and a scalable, reproducible <text,audio> pair generation engine, and extensive evaluation to date of existing TTA models under relation-aware setting. These contributions are central to the "Dataset and Benchmarks" track.

   We agree that new technical methods built upon Aurelius would further advance the field; indeed, we are actively exploring such architectures. But these developments are orthogonal to the core purpose of this paper—which is to define, standardize, and enable research on relation-aware TTA at scale. The absence of a novel method does not diminish the merit of our work. Instead it is fully aligned with the expectations of this track.

Regarding all reviewers' other concerns, we have carefully addressed all of them during the one-one response. We hope the area chair can take our rebuttal and our focus into consideration when making the final decision, and we appreciate you investing time and effort in evaluating our work.

---

### Meta-Review · Area_Chair_kSZ4 · 2026-01-08

**Summary:**

This paper introduces Aurelius, a large-scale benchmark framework with two corpora, AudioEventSet and AudioRelSet, that enables systematic evaluation and development of relation-aware text-to-audio generation. Major concerns form reviewers include (1) potential noise introduced by GPT (2) lack of perceptual quality guarantee (3) distinction from existing datasets (4) the design lacks justification (5)challenging to scale up (6) confusing metrics (7) some writing lacks clarity

**Reviewer Concerns:**

most of the concerns are addressed.

**Reviewer Scores:**

reviewer P1t4 may raise his score

---

### Decision · Program_Chairs · 2026-01-26

Accept (Poster)